

# Correction of CCI cloud data over the Swiss Alps using ground-based radiation measurements

Fanny Jeanneret [1], Giovanni Martucci [2], Simon Pinnock [1], and Alexis Berne [3]

[1]ESA ECSAT, Harwell Campus, Didcot, Oxfordshire, OX11 0FD, United Kingdom
[2]Federal Office of Meteorology and Climatology MeteoSwiss, Ch. de l'Aérologie, Payerne, Switzerland
[3]EPFL, EPFL-ENAC-IIE-LTE, Lausanne, Switzerland

*Correspondence to:* Fanny Jeanneret (Fanny.Jeanneret@meteoswiss.ch)

**Abstract.** The validation of long term cloud datasets retrieved from satellites is challenging due to their worldwide coverage going back as far as the 1980s, among others. A trustworthy reference cannot be found easily at every location and every time. Mountainous regions represent especially a problem since ground-based measurements are sparser. Moreover, as retrievals from passive satellite radiometers are difficult in winter due to the presence of snow on the ground, it is particularly important

to develop new ways to evaluate and to correct satellite datasets over elevated areas.

In winter for ground levels above 1000m (a.s.l.) in Switzerland, the cloud occurrence of the newly-released cloud property datasets of the ESA Climate Change Initiative Cloud_cci project (AVHRR-PM and MODIS-Aqua series) is 132 % to 217 % that of SYNOP observations, corresponding to between 24 % and 54 % of false cloud detections. Furthermore, the overestimations increase with the altitude of the sites and are associated with particular retrieved cloud properties.

In this study, a novel post-processing approach is proposed to reduce the amount of false cloud detections in the satellite datasets. A combination of ground-based downwelling longwave and shortwave radiation and temperature measurements is used to obtain a mask for the cloud cover above 41 locations in Switzerland. An agreement of 85 % is obtained when the cloud cover is compared to surface synoptic observations (90 % within ± 1 okta difference). The obtained cloud mask has been co-located with the satellite observations and a decision tree is trained to automatically detect the overestimations in the

satellite's cloud masks. Cross-validated results show that $62 \pm 13$ % of these overestimations can be identified by the model, reducing the systematic error in the satellite datasets to $4.3 \pm 2.8$ %, at the cost of an increase of $7 \pm 2$ % of missed clouds. Using this model, it is hence possible to significantly improve the cloud detection reliability in elevated areas in the Cloud_cci's AVHRR-PM and MODIS-Aqua products.

## 1   Introduction

Clouds have a major importance in climate: they play a key role in the radiation budget (Trenberth, 2009) and the water cycle, which then impact almost every component of the climatic system, as well as people's everyday life. As the climate changes, cloud properties change as well (Quaas, 2015; Norris et al., 2016; Davies et al., 2017). Detecting and analysing these changes is only possible with high quality datasets spanning several decades. Satellite instruments are the most suitable tools for the global observation of clouds, and scientific effort is increasingly focusing on reprocessing historical records to



extract as much information as possible from them. In 2010, the European Space Agency started the Climate Change Initiative (CCI) programme (Hollmann et al., 2013) to coordinate scientific work towards the production, homogenisation and validation of long-term datasets constructed from different satellite instruments. The CCI project dedicated to clouds, the Cloud_cci (Stengel et al., 2015), is releasing its open-source datasets at the time of writing of this paper. Two of them are of particular

interest here: the dataset based on the Advanced Very High Resolution Radiometer afternoon series (AVHRR-PM, Heidinger et al., 2014), and the dataset based on MODerate-resolution Imaging Spectroradiometer (MODIS) Aqua data. AVHRR-PM has a long time coverage, which gives the opportunity to look for climate change signals. MODIS-Aqua dataset is processed by the Cloud_cci using the same algorithm (CC4CL, Sus et al., 2017; McGarragh et al., 2017) as AVHRR-PM, only with a higher spatial resolution.

In mountainous regions, processes such as elevation-dependent warming (Rangwala and Miller, 2012; Pepin et al., 2015) are documented, suggesting climate might change faster with increasing altitude. Thus, signs of climate change should be easier to observe in elevated areas, either due to larger amplitudes or to earlier appearance. However, mountains are one of the most challenging places for satellite measurements: the accuracy of geolocation is lower over complex terrain, and radiometers measurements are not efficient at discriminating between snow and clouds (Musial et al., 2014). As a consequence, datasets

based only on satellite radiometers have a lower quality in winter in mountainous areas, and this study proposes a way of addressing this limitation by combining ground-based information and machine learning techniques.

Ground-based data have long been used to estimate cloud cover, for instance synoptic observations (Barbaro et al., 1981), measurements of shortwave (Pagès et al., 2003; Long et al., 2006; Martínez-Chico et al., 2011) or longwave radiation (Dürr and Philipona, 2004; Herrmann et al., 2015), as well as active and passive remote sensing instruments such as cloud radars,

cloud lidars and microwave radiometers. The latter ones can indisputably provide accurate measurements of cloud properties like occurrence, altitude and lifetime, but due to their cost they are quite rare and often do not have long historical records. This study hence combines measurements of longwave and shortwave radiation to produce a cloud mask at 41 locations in Switzerland. The new cloud mask covers the period from 1995 to the end of 2014, with different lengths (6.1 ± 4.8 years on average) at different locations, and is validated against synoptic observations for 24 of the 41 stations.

Once the ground-based cloud mask is computed from the radiation data, it is used as labels to train an automated algorithm to detect false cloud measurements in the satellite pixels at the 41 locations. A brief analysis of the types of situations inducing the retrieval algorithm errors is conducted. Time series of cloud properties are also presented, as well as the impact of the removal of points identified as false clouds by the model trained in this study. Then, after investigating the possibilities for spatial extrapolation, the algorithm is applied to every satellite pixel in a defined area to identify false clouds *when* and *where*

no information about the true cloud cover is available. As the focus of this study is on mountainous areas, the geographical zone of interest covers the Swiss Alps. The time frame is 1982-2012 for the AVHRR-PM dataset, and 2002-2014 for MODIS-Aqua.





## 2 Satellite data in the Alps

In this section, an overview of the problems encountered when using two datasets of the Cloud_cci in mountainous regions is presented. The characteristics and limitations of the two datasets are summarised, and cloud occurrences in the European Alps are shown as example. A local validation test is made using 24 ground-based stations in Switzerland, and shows that when

snow is present in the retrieval pixel, cloud amounts are significantly overestimated.

### 2.1 CCI datasets

Two open-source datasets of the Cloud_cci are used: the first one is derived from the AVHRR instruments (Cracknell, 1997) onboard 7 satellites of the National Oceanic and Atmospheric Administration (NOAA). The second dataset is from MODIS instrument (Barnes et al., 1998) onboard Aqua, one of the A-Train satellites of the National Aeronautics and Space Adminis-

tration (NASA). These instruments are passive spectroradiometers measuring top-of-atmosphere radiances at five channels, the so-called AVHRR-heritage channels, centred approximately at 0.6, 0.8, 3.7, 11, and 12 μm. AVHRR on NOAA-16 had a different setup since a 1.6 μm channel was used in daytime instead of the 3.7 μm. For consistency with AVHRR-based datasets, the Cloud_cci MODIS datasets are based on MODIS-Aqua measurements made at these five wavelengths even though the instrument measures at 36 different channels in total.

The level 3U (corresponding to level 2 uncollated data mapped onto a spatial grid), version 002 of the datasets was used (DOIs can be found in the references, under Stengel et al. (2017b) and Stengel et al. (2017c)). The AVHRR-PM dataset covers the years 1982-2012 and has a resolution of 0.05 lat/lon degrees; MODIS-Aqua spans from 2002 to 2012 and is mapped onto a 0.02 lat/lon degrees grid over Europe. All instruments are on polar-orbiting sun-synchronous satellites and overpass locally in early afternoon (around 13:00) and early morning (around 01:00). Since the orbits of the NOAA satellites were allowed to

drift, the local time of each AVHRR observation also drifts by several hours over the lifetime of each satellite (Heidinger et al., 2014).

Cloud properties are retrieved from the satellite-measured radiances using an optimal estimation approach, following the theoretical basis for inverse retrieval methods described in Rodgers (2004). The algorithm, called Community Cloud retrieval for Climate (CC4CL), works in two steps: first, neural networks trained on co-located data from CALIPSO-CALIOP (Winker

et al., 2009) are ran on the radiance temperatures to determine if a cloud is present in the retrieval scheme or not. If so, the retrieval is done both for water and ice clouds using the measured radiances and some ancillary data (sea ice cover, ozone concentration, snow cover, land and sea surface temperature, etc.) coming from ECMWF ERA Interim (Dee et al., 2011). The retrieval converging with the lower cost is kept, yielding five cloud properties: cloud phase, cloud top pressure, cloud optical thickness, cloud effective radius and surface temperature (other variables are computed from there: cloud top height, cloud top

temperature, cloud albedo, liquid and ice water path). The algorithm is described in detail in Sus et al. (2017) and McGarragh et al. (2017).

As detailed in Stengel et al. (2017a), one of the particularities of these datasets is that they include uncertainty estimates at all processing levels. Validation of the Cloud_cci datasets is described in Stapelberg et al. (2017), but as it was done for the





whole Earth, topographic details are not necessarily taken into account. In that report, false alarm rates above 25 % are often seen to occur during daytime above polar snow- and ice-covered surfaces, which might be observed above high-elevated areas as well. When compared to CALIPSO-CALIOP (a satellite lidar instrument), more than 50 % of clouds under 0.15 optical thickness in CALIOP dataset is missing in AVHRR-PM and MODIS-Aqua datasets. Comparison of cloud occurrences with

SYNOP observations (described in the next section) show a good agreement of the seasonal cycles, but overestimations (5 to 10 %) during winter in the Northern hemisphere, particularly mid-latitudes in Europe and Asia.

## 2.2   Visual observations

Surface synoptic (SYNOP) observations are done every 3, 6 or 12 hours at manned meteorological stations: an observer looks at the sky, mentally separating it in an 8-slice pie of which they would be the centre. For each of the eight sky slices, the

observer determines if clouds are present, and if so, estimates their type and altitude. The cloud coverage is hence evaluated in numbers from 0 to 8 called oktas, with an extra value 9 for totally obscured sky (by fog or other meteorological phenomena; these values were discarded). 24 stations in Switzerland are used in this study (a detailed list with operational times can be found in Appendix Table A1).

To ease the comparison with the satellite's binary cloud masks, a limit was set at 0-3 oktas for clear skies and 4-8 oktas

for cloudy skies, which means that only significant amounts of reported clouds will be categorised as cloudy conditions. It is consistent with the different viewing geometries involved, since the human observer might see much further than the satellite pixel's limits when the cloud cover is relatively sparse. Other thresholds were tested, and confirmed that a 3-oktas threshold is an optimal compromise between classifying too many and not enough situations as cloudy. This value is further confirmed by Bojanowski et al. (2014), which use the same threshold.

As the satellites overpass in early afternoon, their observations can be matched to SYNOP observations done at 12 UTC at the 24 stations in Switzerland. At every location, the SYNOP observations are compared with the satellite data falling in the corresponding pixel (approximately 1x1 $km^2$ for MODIS-Aqua and 4x4 $km^2$ for AVHRR-PM). A maximum time difference of 20 minutes was allowed. Some stations might have a larger time difference due to irregular observation times (Dürr and Philipona, 2004), but it was assumed that the observations were in general close enough to 12 UTC. This might cause some

small inconsistencies amongst different stations. Other sources of uncertainties in the comparisons are the subjectivity of human observations and their inevitable variation from one observer to another (Mittermaier, 2012); the scenery effect (Malberg, 1973; Karlsson, 2003; Werkmeister et al., 2015) which increases the difficulty of comparing two different observation geometries, as cloud fractional cover tends to be overestimated by a ground-based observer looking in a slanted way at clouds spread vertically, especially when clouds are low on the horizon; and the detection difference between a human eye limited to the

visible spectra and satellite sensors, which have wider spectral ranges, especially infrared wavelengths.

## 2.3   Satellite cloud mask

A geographical area centred in the Alps was defined as area of interest for this study: it spans from 40 to 51 °N and from 3 to 20 °E (Fig. 1). In this area, MODIS dataset's cloud mask was averaged by season and winter and summer averages are presented



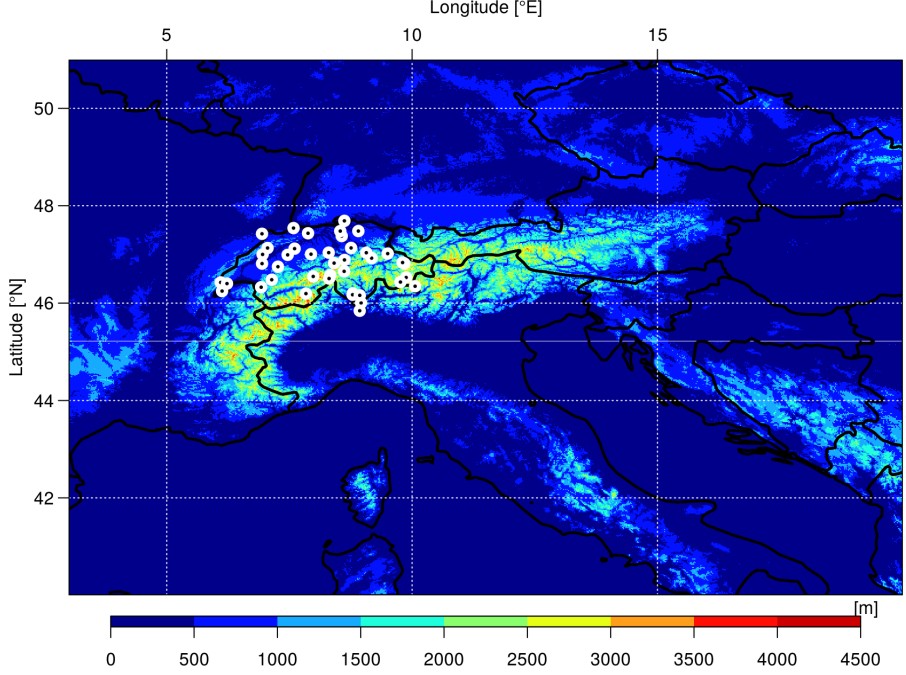

**Figure 1.** Geographical area considered in this study, with ground elevation represented by colors and ground-based stations by white dots.

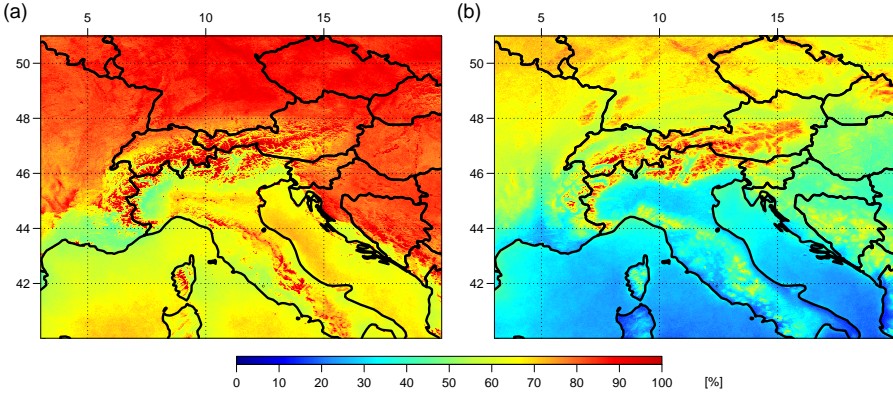

**Figure 2.** Winter (Dec., Jan., Feb.) (a) and summer (June, July, Aug.) (b) averaged cloud occurrences in MODIS-Aqua L3U dataset, years 2003-2014.

in Fig. 2. As can be observed, mountain reliefs are systematically associated with an increase in cloud occurrence, especially in winter. The same pattern is observed when averaging the cloud mask of the AVHRR-PM dataset, but cannot be found in ERA-Interim cloud mask data (not shown).





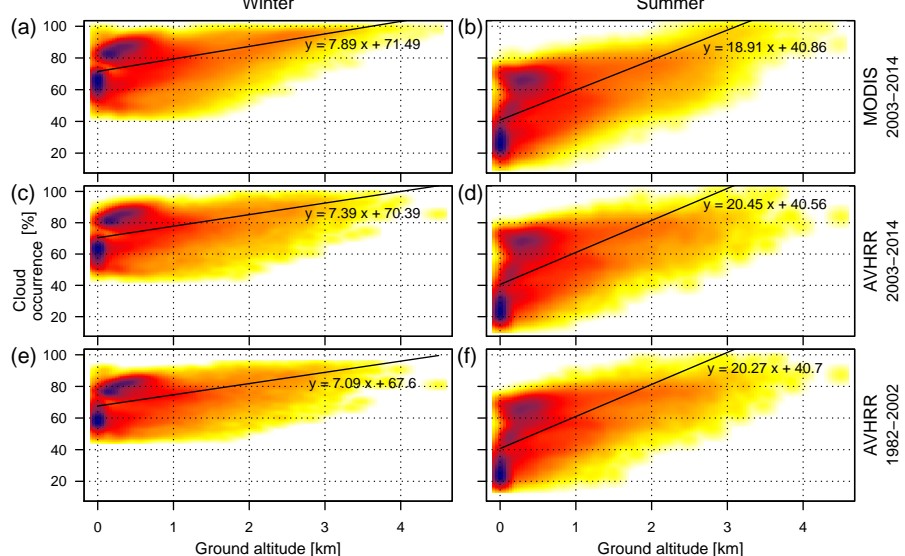

**Figure 3.** Winter (left) and summer (right) cloud occurrences for each pixel in the Alps (area shown in Fig. 2), for MODIS-Aqua 2003-2014 (a and b), AVHRR-PM 2003-2014 (c and d) and the first part of the AVHRR-PM record (1982-2002, e and f). Colours indicate the density of points (the darker it is, the more points there are). A linear regression is drawn and written on each subfigure to illustrate the overall observed relationship between ground altitude and cloud occurence.

The increase of cloud cover with altitude is confirmed independently of the instrument or of the time period considered (Fig. 3). There is a slightly different relationship between ground altitude and cloud occurrence in the two datasets due to their different spatial resolution. In very high areas, the cloud cover is constantly overestimated (it often reaches values larger than 80 % of cloud occurrence) regardless of the season, whereas in lowlands the values found are more consistent and lower in

summer than in winter, which is also observed in a satellite and ground-based instruments intercomparison by Fontana et al. (2013)

In Figure 3, different clusters of points can be observed in winter (two clusters) and in summer (three): they are caused by natural variations of cloud amounts with latitude. The low-occurrence group of points in winter (Fig. 3a,c,e) corresponds to cloud occurrences of satellite pixels above sea and above lowlands south of the Alps (approximately below 46° N) whilst the

10 high-occurrence group contains pixels north of the Massif Central in France, of the Alps and of the Dinaric Alps in Eastern Europe. In summer (Fig. 3b,d,f), cloud amounts get lower and three groups can be identified: the lower one, as in winter, corresponds to pixels above sea or south of the Alps; the middle one contains pixels above lowlands between roughly 46 and 48° N (between the Massif Central and the Alps, and between the Alps and the Dinaric Alps), and the upper group is above lowlands north of the Alps (over 48° N). Winter retrievals of AVHRR-PM 2003-2014 (Fig. 3c) have lower cloud occurrences

(approx. −3 %) than those of AVHRR-PM 1982-2002 (Fig. 3e). As discussed later on, the satellites time drifting might have had an impact on the values.





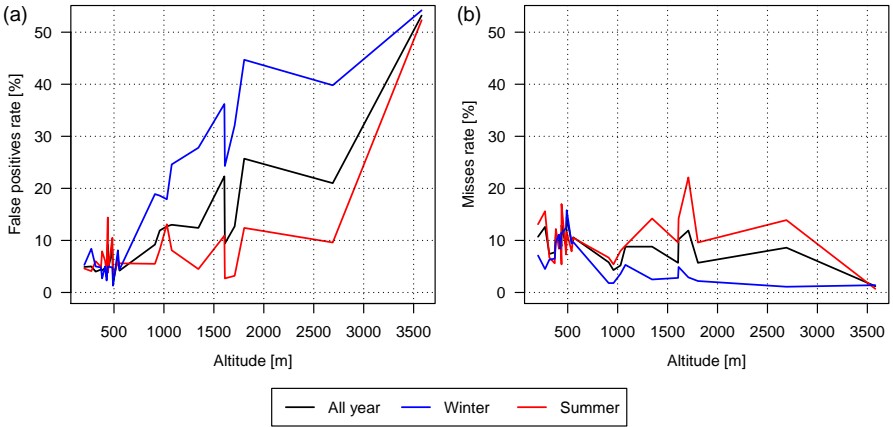

**Figure 4.** MODIS-Aqua cloud mask compared to 24 SYNOP stations in Switzerland. The SYNOP-MODIS overlaps varies from one station to another, from 3.3 years to the whole MODIS record (12 years). The satellite mask false positives (a) correspond to when the satellite dataset contains a cloud, but the ground observer reports a value of less than 4 oktas of clouds. The misses (b) are the opposite disagreement between the two.

When comparing the cloud mask of MODIS-Aqua to SYNOP observations (Fig. 4), extreme values are seen at the Jungfraujoch station (3580m above sea level), where clouds are reported 98 % of the time in the satellites datasets, corresponding to 53 % of false positives. This rate (Fig. 4a) indeed increases with altitude, more drastically in winter and consistently with the spatial pattern observed before. Night satellite measurements are not exempt from these overestimations, although fewer

SYNOP data are available as reference (elevated stations are not manned at night). The decrease of the misses rate with altitude (Fig. 4b), especially in winter, is directly related to the high overestimation rate at these locations. Except for that, the misses rate observed is relatively steady with altitude. It shows a systematic bias of 5-10 %, most likely caused by the different geometries involved in the comparison. For instance, ground observers might see much further than the boundary of the satellite pixel in which they stand, especially in locations without surrounding relief blocking the view. Considering also the limitations

detailed in Subsection 2.2, the cloud masks are overall considered as agreeing under 1000m.

These results suggest that the presence of snow on the ground, in winter and in summer in high-altitude locations, tricks the satellite retrieval algorithm into detecting more clouds than it should. This is consistent with the Appendix of Stengel et al. (2017a), which mentions that the known limitations of these satellite datasets include "*shortcomings in cloud detection and optical property retrievals in regions with high surface reflection of solar radiation*". Snow reflectance leads to top-

of-atmosphere radiances very similar to water and/or ice clouds in different channels. Some methods can be used to help distinguish between them (see for instance Musial et al. (2014)). A widely used solution is to complement spectral data with ancillary data: CC4CL, the retrieval producing the satellite datasets shown here, is indeed based on the snow mask of ERA-Interim. However, given the issues observed here, the spatial resolution (0.7 °lat/lon, MODIS-Aqua being 0.02 ° and AVHRR 0.05 °), quality or use of the ancillary data might be insufficient.





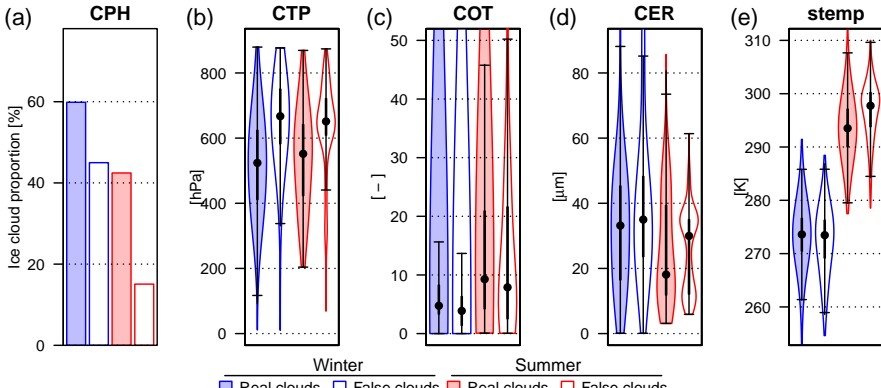

**Figure 5.** Distributions of values of five cloud properties (phase (CPH), top pressure (CTP), optical thickness (COT), effective radius (CER), surface temperature (stemp)) retrieved from MODIS-Aqua in winter and summer at 9 stations located above 1000m of altitude, for actual clouds and falsely detected ones (using SYNOP observations as reference).

## 2.4 Satellite cloud properties

As the cloud property retrieval from satellite-measured radiances is run after the cloud mask computation by the artificial neural networks, any mistake at this first stage carries on to the next. Retrieval of cloud properties on an absence of cloud often leads to values differing from those associated with actual clouds. For instance, false clouds are more often of liquid phase than ice (Fig. 5a), are significantly closer to the ground (higher top pressure, 5b), have lower cloud optical thickness but higher cloud effective radius (5c and d), and have higher surface temperatures (5e). A surprisingly high mode can also be seen in the effective radius distribution (5d) of false clouds in summer, corresponding supposedly to ice particles. These observations are consistent with a retrieval influenced by the presence of snow: these false clouds are lower, warmer and with larger particles than actual clouds. In summary, cloud mask errors have an important impact on the retrieved cloud properties as well, and identifying such cases before any in-depth analysis is very important.

## 3 Radiation cloud mask

The previous section presented briefly how the CC4CL satellite retrieval overestimates cloud amounts above elevated areas. The next two sections propose a solution to handle this issue: first, a new binary cloud mask is defined. The combination of several ground-based observations provides insight about the cloud cover at 41 locations in Switzerland. This allows the use of a larger amount of locations than the SYNOP stations, especially with more data in elevated areas and without potential issues regarding the subjectivity of SYNOP observations. Subsequently, this cloud mask is used as reference to train a model for the automated detection of false clouds in the satellite datasets.





## 3.1 Ground-based data

Downwelling longwave and shortwave radiations as well as 2-metre ground temperature are used in this study to get an estimation of the local cloud cover. Longwave and shortwave downwelling radiation is measured by pyrgeometers, respectively pyranometers, which consist in a thermopile sensor and a temperature sensor under a small dome. The pyrgeometer's spectral

band is 4.5 - 42 µm, whereas the pyranometer's is 0.3 - 3 µm. Both instruments have a field of view close to an ideal 180 degrees (the exact values depend on the instrument's quality), and each measurement is weighted by the cosine of the incidence angle, giving more importance to radiation at angles close to the zenith.

All measurements were converted to 10-minute averages. Information about the measurement setup and data preprocessing can be found in the work by Dürr and Philipona (2004). Of the 41 stations used in this study, 37 are part of the SwissMetNet

network (Suter et al., 2006) operated by the Swiss weather office MeteoSwiss, and 4 are part of the Alpine Surface Radiation Budget (ASBR) network (Marty et al., 2002). The pyrgeometers used in the SwissMetNet network are of type CG4 and CGR4 from Kipp & Zonen, with a declared uncertainty of 3 %. Older measurements on the ASRB network have been taken by modified Eppley PIR pyrgeometer, which have an observed uncertainty of 3 $\mathrm{Wm^{-2}}$ (Marty et al., 2002). The pyranometers are mostly CM21 from Kipp & Zonen (2 % uncertainty) and a few SPN1 from Delta-T (5 % or 10 $\mathrm{Wm^{-2}}$). A detailed list of

stations including operation times can be found in Appendix Table A1.

## 3.2 Topographic data

Topographic information come from the freely available Global Digital Elevation Model (GDEM) ASTER version 2 (Tachikawa et al. (2011); ASTER GDEM is a product of METI and NASA). This GDEM has a spatial resolution of 1 arc-second (approximately 30 metres at the equator).

## 3.3 Method for the ground-based cloud mask

The method described here combines two different types of radiation to estimate the state of the sky at 41 locations, with a 10-minute temporal resolution.

Ground-based longwave measurements have been used in various ways to estimate cloud cover (e.g., Dürr and Philipona, 2004; Dupont et al., 2008; Viúdez-Mora et al., 2009). The method used here is inspired by the work of Herrmann et al. (2015)

and consists in converting downwelling longwave measurements ($L$, $\mathrm{Wm^{-2}}$) into estimated sky temperatures ($T_{sky}$, K), then comparing them to ground-based (2 metres) temperatures. As the radiation emitted by a cloud is comparable to that of a black body at the same temperature, Stefan-Boltzmann law is used for the conversion

$$T_{sky} = \left(\frac{L}{\epsilon\sigma}\right)^{\frac{1}{4}} \tag{1}$$

where $\sigma$ is Stefan-Boltzmann constant ($5.67 \cdot 10^{-8}$ $\mathrm{Wm^{-2}K^{-4}}$) and the atmospheric emissivity $\epsilon$ is approximated to 1.

Figure 6 shows cluster plots of the estimated sky temperature versus the measured ground temperature. Clear sky estimates always cluster at low temperatures (being a mixture of atmospheric and cosmic background temperatures), whereas cloud





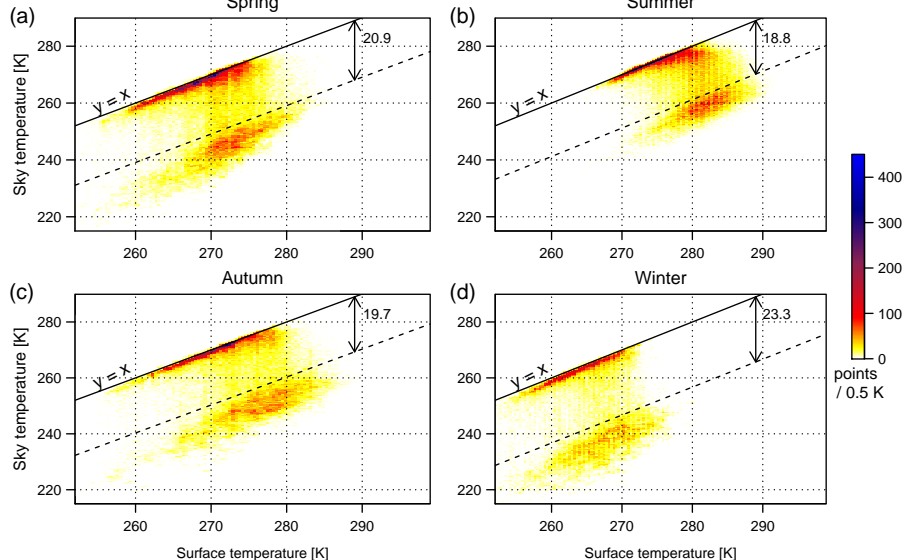

**Figure 6.** Comparison of daytime sky and ground temperatures at Weissfluhjoch station (2690m) in Switzerland, per season, using data from 1994 to 2010. The upper cluster corresponds to in-cloud and overcast conditions, whilst the lower one corresponds to clear-sky times. The points in-between are partially cloudy conditions. The dashed line corresponds to the automatically detected threshold under which values are considered as being part of the lower cluster.

base temperatures are significantly higher (representative of tropospheric temperatures at the cloud height). Scattered cloud conditions are characterised by values falling between the two main clusters.

The detection of the lower cluster border is done in five steps. First, the differences between ground and sky temperatures are calculated. Then, the density distribution of these values is computed (black curve in Fig. 7) and smoothed (red curve).

One way to find the cluster border is to look for a strong density increase in this distribution, so in the third step, the derivative (blue curve in Fig. 7) is computed. As the upper cluster is spread over several kelvin degrees, only temperature differences larger than 5 K can correspond to the lower cluster. The fourth step consists in looking for the maximum of the derivative for differences over 5 K. Lastly, the cluster border is set at the minimal temperature difference so that half this maximum is reached (yellow vertical line in Fig. 7). The sensitivity of this value was analysed and showed that a change of $\pm$ 20 % had a

very limited impact on the size of the lower cluster. It did not have a significant effect on the correlation of the resulting cloud mask with SYNOP observations either.

Due to the daily temperature cycle, days and nights are clustered separately to ensure that accurate lower cluster limits are obtained. Local sunrise and sunset times are used as time limits.

Different applications of this method are shown in Fig. 7, for daytime conditions in winter and summer. One station with

a long data record (Weissfluhjoch, 16 years) is compared to another one with a very short record (Segl-Maria, one year). As





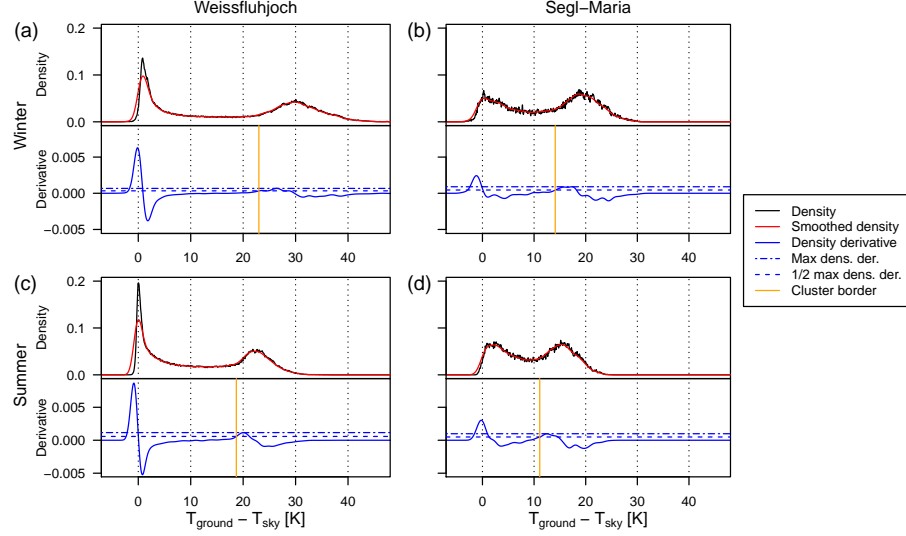

**Figure 7.** Detection of the longwave lower cluster on daytime data, at two different stations (left column, Weissfluhjoch; right column, Segl-Maria, one year of measurements), for winter (first row) and summer (second row) seasons.

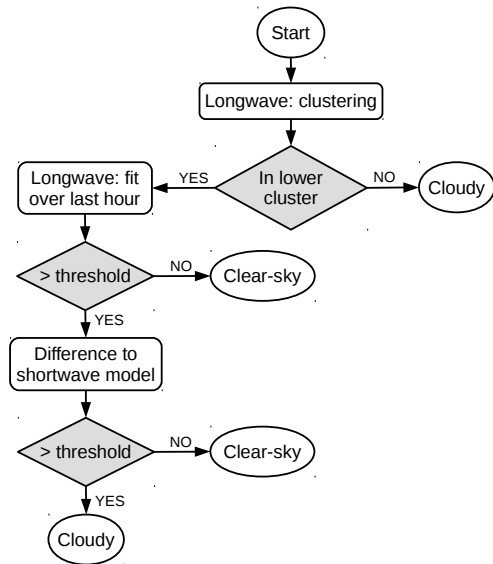

**Figure 8.** Cloud mask algorithm

can be seen, the main advantage of detecting the cluster's border as a strong density increase is the excellent adaptability to different amounts of data as well as to different cluster shapes.





After this clustering step, two other criteria are combined to discriminate partially cloudy conditions from clear-sky ones (Fig. 8). The stability of the longwave measurements over the preceding hour is used as a sign of partial cloud cover, as in Dürr and Philipona (2004). However, a cloud is detected only if this criterion reaches a given threshold at the same time as the difference between estimated and measured shortwave radiation exceeds another threshold. Exploration of the parameter space in 2D has shown that this second criterion improves the classification, and that the threshold values are not particularly sensitive. The stability of the longwave measurements over the preceding hour is computed as the root-mean-square deviation between the values and a linear fit applied to them (the threshold in the algorithm is set to $1.75\ \mathrm{Wm}^{-2}$).

The shortwave criterion, $C$, is defined as a weighted sum of relative differences between the measured and estimated global radiation over the preceding hour (the threshold is set to 0.15 and is dimensionless). The weights are larger close to the time of interest $t$:

$$C = \frac{1}{28} \sum_{i=0}^{60} \frac{(70-i)}{10} \cdot \left| \frac{S_{e,t-i} - S_{m,t-i}}{S_{e,t-i}} \right| \qquad (2)$$

where $i$ is a time index varying between 0 and 60 minutes by steps of 10 minutes, and $S_{m,t}$ and $S_{e,t}$ are respectively the measured and estimated global radiation, in $\mathrm{Wm}^{-2}$, at time $t$.

The incoming global radiation $S_{e,t}$ is estimated using a simple model described in Sun et al. (2013), the model SW1 in their paper:

$$S_{e,t} = \tau \frac{S_0}{d_t^2} \cdot cos(\theta_t) \qquad (3)$$

where $\tau$ is the atmospheric transmittance (dimensionless), $S_0$ the solar constant ($1367\ \mathrm{Wm}^{-2}$), $d_t$ the Earth-Sun distance (in astronomical unit, AU) and $\theta_t$ the solar zenith angle (in radians) at time $t$. The particularity of this model is that the atmospheric transmittance $\tau$ is approximated as a linear function of the altitude, following Tasumi et al. (2000).

The shading effect of topography is taken into account due to its significant effect on radiation in mountainous areas (Lai et al., 2010). The shading angle $H$ is defined as the minimal elevation above horizon required to see the sky above the surroundings. When the sun is at a given azimuth $\phi$, if its elevation is below $H(\phi)$ then the estimated radiation is set to zero. The shading angles were computed at each station using ASTER GDEM, taking into account surroundings up to a distance of 0.5 lat/lon degrees (55km/38km at 46°N), with a resolution of approximately 1 arc-second (15m/10m at 46°N).

## 3.4 Results

The obtained cloud mask was validated against SYNOP observations, with a time difference of at most 10 minutes. The percentages of misclassified clouds per okta is shown in Fig. 9. The largest source of error comes from the transformation of the SYNOP observation into a binary threshold: it is difficult for the radiation cloud mask algorithm to follow this strict threshold when the distinction between 3 and 4 oktas is quite subjective. The cloud mask is hence accurate 85.4 % of the time, and this value reaches 90.3 % if both clear and cloudy classifications are allowed for 3- and 4-okta observations. With this 1-okta difference allowed, the probability of correctly detecting clouds is of 87.6 % and the probability of false detections of





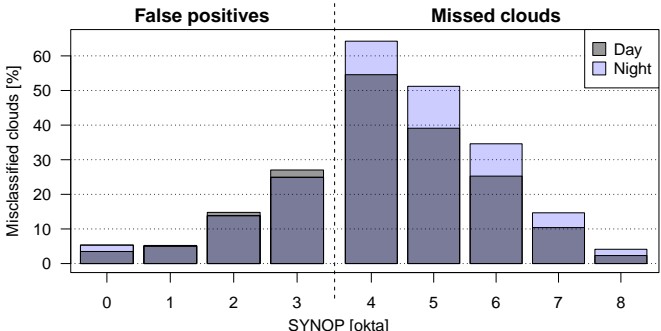

**Figure 9.** Distribution of the ground-based cloud mask errors as a function of SYNOP observations at 21 stations. The vertical dashed line represents the binary threshold applied to the SYNOP observations.

5.6 %. This is consistent with Fig. 9, because more clear (0-1 okta) and cloudy (7-8) conditions are recorded than partially ones. A bias of 0.913 oktas is present, confirming a tendency to miss some clouds. Thin and high clouds cause only minor perturbations in the radiation measurements compared to the clear sky conditions, and for that are more likely to be missed. A larger amount of clouds are missed at night due to the lack of shortwave information. On the contrary, the negligible difference

between false positives between day and night shows that a sparse cloud cover missed by the radiation instruments at night is most likely missed also by the human observer. The results' consistency is good among the different stations (the averaged accuracy is $91 \pm 2\%$), the lowest accuracy being at the Jungfraujoch (86 %), which is on a mountain pass where clouds can be observed under the observatory.

Even though the accuracy of the cloud mask presented here is somehow limited with partially cloudy cases, these results are

10 of great value for validation processes in need of stable long-term references, such as satellite instruments. The results further suggest that this method could potentially be used outside the geographical area evaluated here, as the clustering method should adapt automatically to different climatic conditions.

## 4   Automated detection of false clouds

In this last section, a model is trained to automatically detect false positives in the satellite datasets. Using the radiation

cloud mask as reference, it combines several variables retrieved from satellite-measured radiances with information about the topography and time of the retrieval. The possibility of using the model at other times than those used as training is considered, and the model is applied to long satellite time series. Similarly, its ability to extrapolate at other locations than where it was trained is evaluated and maps of its effects are discussed.





## 4.1 Methods

A decision tree was trained to automate the identification of false clouds in the satellite datasets. The model's inputs are the five variables retrieved from the satellite radiances by the CC4CL algorithm (cloud phase, cloud optical thickness, cloud top pressure, cloud effective radius and surface temperature), as well as the ground altitude, the standard deviation of the surrounding ground altitudes, and a time variable. Time is represented as a sinusoidal function with a period of one year, peaking on the 15$^{th}$ of January (+1) and on the 15$^{th}$ of July (-1). The standard deviation of the surroundings is computed within a radius of 3 km on the 30-metre GDEM. Using these variables, the model predicts if the sky actually contains a cloud or not. The radiation cloud mask defined in Section 3 is used as reference for training. The training is done by 10-fold cross-validation with random sampling. Testing metrics are computed over the ten models obtained, and only the best one is then validated against SYNOP observations. After this validation, the structure of the model is discussed and some groups of points are defined. Focusing on these groups allows analysing the whole dimension space without considering each cloud property or each point one by one. Performances of the model are tested on each of these groups and allow some potential weaknesses to be identified.

Once validated, the decision tree is applied to the two satellite datasets presented at the beginning of this study. First, the effect of the model on times series of cloud properties is discussed. Then, leave-one-out validation is done to assess if the model can be applied at locations where it was not trained, and what kind of results can be expected in these circumstances. Leave-one-out validation consists in training several models, each with a training set composed of all but one station. Testing is done on this last station, and all the testing results are regrouped. Important information about the model weaknesses can be deduced from where the model had difficulties to adapt without training. It provides an overall idea about how the model will perform at locations where no reference data is available. Once this is done, the model is applied to a larger geographical area and the results are discussed as another insight on the model's strengths and weaknesses.

## 4.2 Analysis and validation of the model

The model obtained at the end of the training process is a large decision tree drawn in Fig. 10. After looking into the overall results, the groups of points circled in this figure are analysed more in detail.

Overall, the test metrics give a probability of 82.6 % of detecting false positives, and of 10.9 % of false detections. They are computed using the radiation cloud mask as reference, and averaged over the 10 tests of the cross-validation. When validated against SYNOP observations, results show that in winter above elevated areas, where most of the satellite false cloud detections happen, $73 \pm 12$ % of errors are identified (Table 1). The amount of missing clouds in these conditions is increased by $10 \pm 4$ %, whereas lower values are found in all other conditions. In summer, around 45 % of the overestimations are detected, with quite large differences between the stations, but with no significant link to the station's altitude. Globally, $62 \pm 13$ % of the cloud mask overestimations are detected, reducing the systematic false positive error from $14.4 \pm 15.5$ to $4.3 \pm 2.8$ % but increasing the missed clouds from $8.7 \pm 3.5$ to $15.6 \pm 2.1$ %.





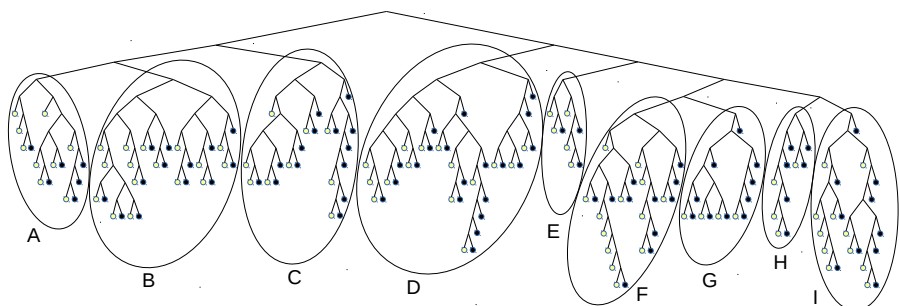

**Figure 10.** Structure of the model. Small circles correspond to the tree final nodes (also called leaves): light ones are classified as clear-sky, dark ones as cloudy. The parts circled and labelled with letters corresponds to the groups of points defined in Table 2.

**Table 1.** Evaluation of the model's effects on MODIS dataset using SYNOP data as reference, at 20 SYNOP locations (7 above 1000m, 13 below; 4 stations with limited overlap between SYNOP and the ground-based radiation measurements were excluded). The two left columns contain the percentage of false positives in the satellite data that were correctly identified by the model. The two right ones contain the percentage of clouds identified in the satellite data which are considered false by the model, even though they are actual clouds according to binarised SYNOP observations.

|  | False positives correctly identified by the model [%] | | Clouds wrongly identified by the model [%] | |
|  | Winter | Summer | Winter | Summer |
| --- | --- | --- | --- | --- |
| Above 1000m | $73 \pm 12$ | $43 \pm 24$ | $10 \pm 4$ | $5 \pm 3$ |
| Below 1000m | $33 \pm 24$ | $47 \pm 17$ | $3 \pm 3$ | $5 \pm 5$ |

As proposed in Fig. 10, groups of points can be delineated in the tree and are characterised by the criteria detailed in Table 2. All of these groups are separated in subgroups that are not described here. As can be seen in the table, the cloud optical thickness (COT) is one of the most important variables in this model. Groups A and B, characterised by a small COT and which contain a large amount of well-classified points, most likely also contain a non-negligible amount of cirrus clouds, too thin to be seen

5 by the reference cloud mask produced in this study. Group D is the only one containing only clouds located above mountains, and its high detection percentage confirms the efficiency of the model. Groups C, G, H, I are all characterised by low model performances. Group C can be understood as a small group of points scattered in the dimension space and which did not trigger a particular response of the model. The others are mainly composed of large COT, and group G probably corresponds to liquid phase clouds (CTP under 627 hPa and CER under 22 μm).

10 As can be observed, groups of points that seem related (for instance, F and G) can be understood very differently by the model, which suggests that a more complex model could be necessary to catch subtle differences between false and real clouds, and improve the results.



**Table 2.** Main branches of the tree. The letters in the first column correspond to the groups circled in Fig. 10. The second column states the amount of false positives (FP) in the satellite data falling into each branch, the third one how many of them were correctly identified by the model (IFP: Identified False Positives). The remaining columns describe the parameters and thresholds leading to each branch. COT: cloud optical thickness, CTP: cloud top pressure, CER: cloud effective radius.

| Group | FP [%] | IFP [%] | Criteria | | | |
|---|---|---|---|---|---|---|
| A | 10.5 | 87 | $COT < 1.47$ | time: not in winter (between mid-March and mid-November) | | |
| B | 13.0 | 73 | $COT < 1.47$ | time: in winter (between mid-November and mid-March) | | |
| C | 6.5 | 37 | $1.47 \leq COT < 2.36$ | | | |
| D | 15.1 | 80 | $2.36 \leq COT$ | ground altitude $\geq 2988$ m | | |
| E | 3.9 | 92 | $2.36 \leq COT$ | ground altitude $< 2988$ m | $CTP < 215$ hPa | |
| F | 9.2 | 82 | $2.36 \leq COT$ | ground altitude $< 2988$ m | $627 \leq CTP$ | $CER \geq 22$ μm |
| G | 17.8 | 12 | $2.36 \leq COT$ | ground altitude $< 2988$ m | $627 \leq CTP$ | $CER < 22$ μm |
| H | 4.5 | 18 | $2.36 \leq COT < 4.03$ | ground altitude $< 2988$ m | $215 \leq CTP < 627$ hPa | |
| I | 19.4 | 10 | $4.03 \leq COT$ | ground altitude $< 2988$ m | $215 \leq CTP < 627$ hPa | |

## 4.3 Result of filtering the satellite dataset using the decision tree model

Having looked at the limitations of the model and at the expected results, the decision tree was then applied to the satellite-derived cloud property time series. The pixels corresponding to 9 stations above 1000m were extracted from the satellite dataset, and false clouds were detected and removed using the decision tree. The main cloud properties were observed before

5 and after removal of these false positives, and are averaged per year and per month in Fig. 11. Since the model is trained on MODIS-Aqua dataset only (2003-2014), a temporal extrapolation is done to the whole NOAA AVHRR-PM time series (1982-2014).

After removal of the points identified by the model as likely not to be clouds, the cloud fractional cover (CFC) is lower. As expected, the points removed were more in winter than in summer, at higher altitudes, with larger optical thickness and

10 smaller effective radius. This is highly consistent with the changes observed when removing from MODIS-Aqua dataset the clouds not observed by a human observer (Fig. 5). The difference between MODIS-Aqua and the MODIS-Aqua corrected time series seems constant over time, and suggests that the effects of the model are temporally consistent. Over the years 2003-2014 the two satellite datasets agree very well, except for a small offset caused by the difference of spatial resolutions. Since the AVHRR series used here is not homogenized regarding e.g. drifts in overpass time between different NOAA satellites (Stengel

et al., 2017a), its behaviour is not stable over time. This can be seen over the years 1982-2002, near the end of each NOAA satellite, where values diverge progressively from their average. Some of these values are not expected by the model and tend to be removed, causing very low CFC values for several winters (at the end of 1983, 1984, 1987, etc.). As the satellites were drifting in time (up to 3h30 for NOAA 11), it suggests the need of correcting the cloud properties for the cloud diurnal cycle before applying a model using them.





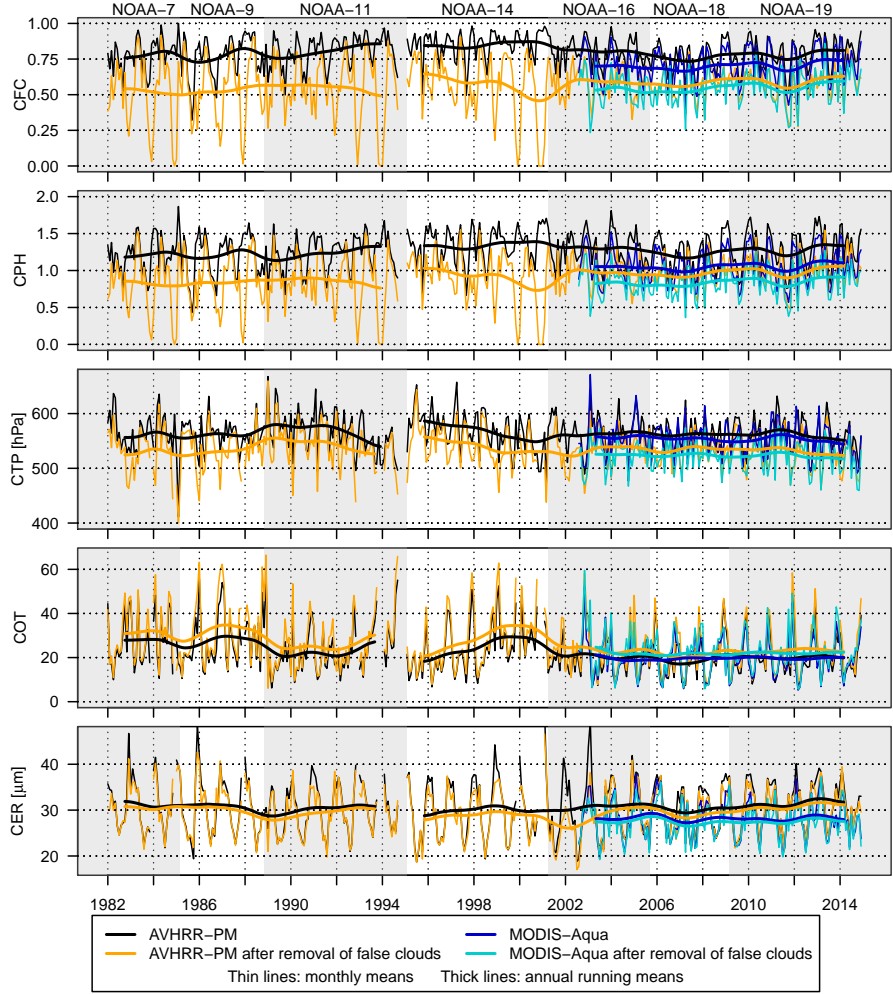

**Figure 11.** Time series of monthly means (thin lines) and annual moving averages (thick lines) of cloud properties, before and after removing points flagged by the model as potential false positives. 9 SYNOP stations located above 1000m are averaged together. CFC stands for cloud fractional cover, CPH for cloud phase (0: no cloud, 1: liquid phase, 2: ice phase), CTP for cloud top pressure, COT for cloud optical thickness and CER for cloud effective radius.

When compared to the latitude-weighted 60°S-60°N time series in Stengel et al. (2017a), the time series in Fig. 11 have wider seasonal amplitudes. The satellite's drifting in time also has a larger impact on the values. Both are related to the size of the areas over which the data are averaged. The proportion of cloud ice particles is significantly higher in the time series in Switzerland than in Stengel et al. (2017a), even in low areas (results not shown), and after applying the model. Similarly, the cloud optical thickness is approximately twice larger in the time series presented here than in Stengel et al. (2017a). The cloud effective radius decrease from mid-2001 to mid-2003 is due to the change of channel on NOAA-16 (Heidinger et al., 2014), and can be observed in Stengel et al. (2017a) as an increase of effective radius. The channel 3.7 μm was switched to





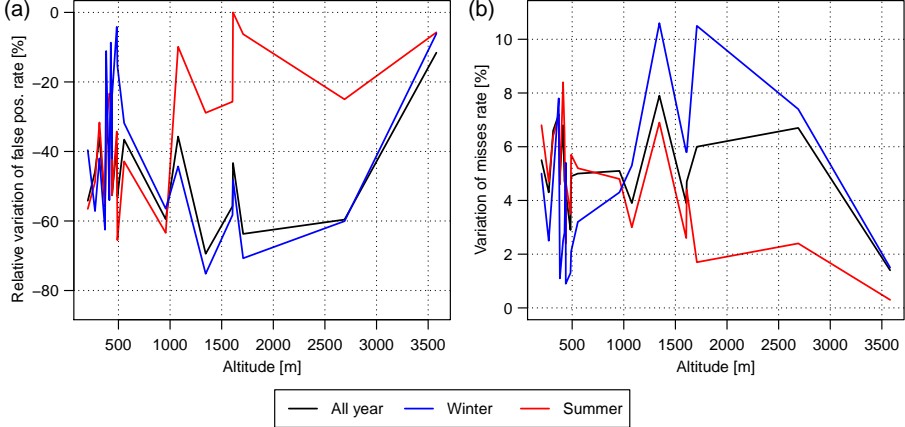

**Figure 12.** For each SYNOP station, a model was trained on all stations except one (leave-one-out validation). The figure shows the effect each model had on the rates of false positives (a, relative percentages) and missing clouds (b, absolute percentages) at the station left out, using SYNOP observations as reference.

1.6 μm, which seems to trigger the model. On this same property, a bias can be observed on values retrieved from NOAA-19, which seem to have an increasing difference with MODIS-Aqua. This can also be seen in Stengel et al. (2017a) and might be an instrumental bias.

### 4.4 Leave-one-out validation

The leave-one-out validation results in Fig. 12 suggest that the model can reliably be generalized in space, especially for elevated areas in winter. The highest station of the dataset (Jungfraujoch) is an extreme case in this dataset, and cannot be well understood by the model if it is not part of the training set. Although good, the performance above elevated areas is lower in this validation than the values obtained previously on a slightly larger dataset (Table 1, errors correctly identified above 1000m). This suggests that increasing the amount of stations would be beneficial. However, even without increasing the number of

stations, significant results (more than 50 % of the satellite cloud mask overestimations identified) can still be expected.

    In this Figure 12, one can also observe that the detection of false positives in the satellite data is less efficient above elevated areas in summer, whereas in lower areas there is no significant seasonal difference. This might be a sign that seasonal differences in mountains are not fully understood by the model, probably due to the lower amount of stations above 1000m than below. As winters contain a much larger amount of false positives than summers, false positives in summer in high altitudes

end up being poorly represented in the dataset. Using a probabilistic approach taking into account the different amount of points in each condition might help reducing the model's skewness in favour of the detection of the most recurring problems.

    No link seems to exist between the ability of the model to generalize to a station and the complexity of the station's surrounding, suggesting that the model already gets the most out of this variable. For stations below 1000m, a moderate negative correlation (-0.46) was found between the average relative increase of missing clouds caused by the model and the distance



between a station and the center of its corresponding satellite pixel. A similar correlation (-0.43) was observed between this distance and the ability of the model to detect false positives in winter. This suggests that lower performance is partially caused by a spatial offset between the satellite viewing scene and the ground-based one. A solution out of the scope of this study would be to look more in detail into the geometry of the viewing scenes, and maybe to combine several satellite pixels in a weighted

mean for comparison with a ground-based station.

### 4.5 Larger scale model application

Lastly, the model was applied on a large area and maps of the effect on the cloud coverage were produced (Fig. 13). They confirm that even at locations outside the training data, the model reduces cloud occurrences to more reasonable values above elevated areas, especially in winter (Fig. 13a). The systematic removal of 7 percent of the clouds (identified as false positives

even though they were most likely real clouds) is not restricted to a specific area.

With a training set made of stations located only in Switzerland, the results appear very consistent above land also outside this area. Considering the model output over a wider area however illustrates its limitations, mainly above sea and lakes at every season but winter. Especially in summer, clouds are detected as false positives above water bodies, as their surface temperature is lower than the one of the surrounding ground. For instance, a blue spot can be seen North of Italy in the Po valley (approx.

45.2 °N, 8.5 °E), where rice cultures are flooded during summer months, which significantly lowers the ground temperature. This confirms the importance that the model gives to the surface temperature, and suggests that either the model should be applied only above land, or it should be aware of the land/sea difference (for instance, adding a land/sea mask would be an easy step, but finding radiation measurements above sea for training might be difficult).

As applying a decision tree on data is a very quick process, this model is a simple solution to remove a significant amount

of issues over elevated areas, for the reasonable cost of decreasing slightly and homogeneously the amount of clouds.

### 5 Conclusions

Two satellite datasets of ESA's CCI on clouds were seen to overestimate the cloud cover above elevated areas. MODIS-Aqua and AVHRR-PM can contain up to 54 % of false cloud detections in winter in mountainous areas (above ground with an elevation higher than 1000m). These cloud mask errors also have an important impact on the cloud properties, as retrievals on

missing clouds often have unexpected values. Identifying them prior any detailed analysis is a necessary step.

Using longwave and shortwave radiation measurements at 41 stations in Switzerland, a binary cloud mask was defined. It is tailored to each station and each season thanks to a simple automated clustering of the longwave data. Shortwave data and a second longwave criterion are then used to provide more insight in partially cloudy cases. Validation against SYNOP shows that the model has a probability of 87.6 % of detecting cloudy skies using this combination of ground-based information, and

a probability of 5.6 % of false detections.

This ground-based cloud mask was then used as reference to train a model for the detection of false clouds in the satellite datasets. The model's input contains variables such as the satellite-retrieved cloud properties, ground and time information.

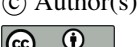


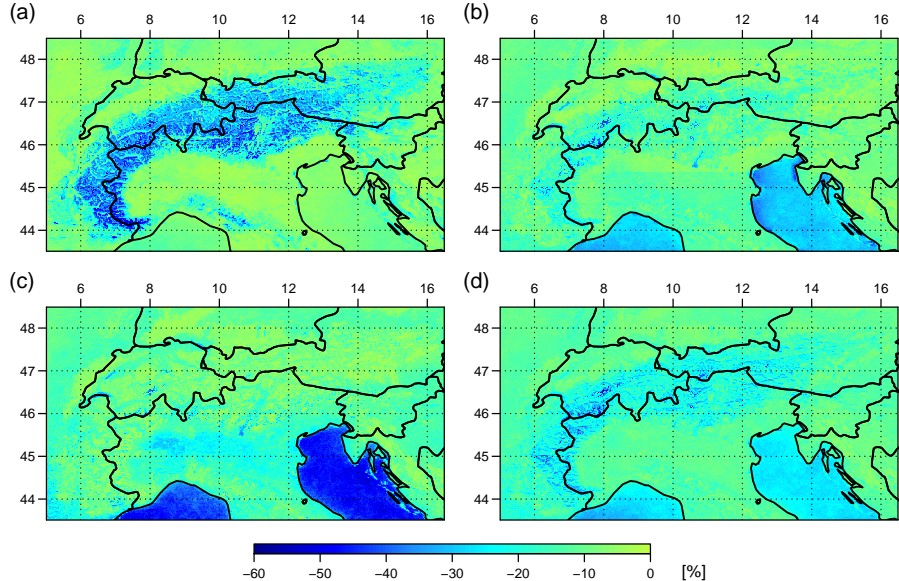

**Figure 13.** Effect of the model on cloud occurrences, in percent relative to the initial cloud occurrences. The results are averaged over MODIS-Aqua dataset (2003-2014) by season: winter (a), spring (b), summer (c) and autumn (d). The seasons are defined as Dec.Jan.Feb., Mar.Apr.May, JuneJulyAug., Sep.Oct.Nov.

In the Swiss Alps area, the use of the decision tree model as a quality filter permitted the rejection of 62 % of the false cloud detections in the satellite cloud property dataset, with the limitation of causing the removal of 7 % of real clouds in the process. This made a significant improvement to the quality of the satellite cloud property data set over this area. These results are interesting for any application where one can afford reducing the amount of data in order to increase its quality. Improved results might be obtained by using a probabilistic approach, likely to allow under-represented categories to be better understood. A higher number of elevated stations could also be beneficial. Expanding the study area to other latitudes using either already computed cloud masks, or data from the worldwide Baseline Surface Radiation Network (BSRN Ohmura et al., 1998) for instance, would be an expected follow-up.

Considering how time- and resources-consuming is the computation of large satellite datasets, fast post-processing algorithms such as the one proposed in this study are likely to be interesting solutions as more and more data are available. Moreover, as demonstrated here, having several datasets produced by the same retrieval algorithm is a great asset as it allows them to be post-processed in the same manner.



# Appendix A: List of ground-based stations

*Competing interests.* The authors declare that they have no conflict of interest.

*Acknowledgements.* This work was funded by the European Space Agency within the framework of the Climate Change Initiative. Most of the work was carried out at ECSAT (ESA). We would like to thank MeteoSwiss for providing the radiation and synoptic data, and the

5    scientists of the Remote Sensing Group at Rutherford Appleton Laboratory for their support and advice.





**Table A1.** Location and data coverage of the 41 ground-based stations used in this study. The length of overlap with MODIS is not specified when the whole record overlaps temporally with MODIS time range (2002-08-01 to 2014-12-31).

| Location | Altitude [m] | Latitude [deg. N] | Longitude [deg. E] | Radiation | | | SYNOP | | |
|---|---|---|---|---|---|---|---|---|---|
| | | | | Start | End | Length [years] (MODIS overlap) | Start | End | Length [years] (MODIS overlap) |
| Aadorf | 539 | 47.47987 | 8.90487 | 2006-10 | ✓ | 8.2 | 1980-01 | 2007-02 | 27.2 (4.6) |
| Aigle | 381 | 46.32664 | 6.92442 | 2005-09 | ✓ | 9.3 | 1981-02 | ✓ | 33.9 (12.4) |
| Altdorf | 438 | 46.88702 | 8.62180 | 2008-12 | ✓ | 6.1 | 1980-01 | ✓ | 35.0 (12.4) |
| Bad Ragaz | 496 | 47.01662 | 9.50257 | 2012-02 | ✓ | 2.9 | - | - | - |
| Basel | 316 | 47.54103 | 7.58356 | 2009-12 | ✓ | 5.1 | 1980-01 | ✓ | 35.0 (12.4) |
| Bern | 552 | 46.99074 | 7.46400 | 2009-09 | ✓ | 5.3 | - | - | - |
| Chasseral | 1599 | 47.13176 | 7.05439 | 2010-10 | ✓ | 4.2 | - | - | - |
| Château-d'Oex | 1029 | 46.47981 | 7.13964 | 2012-02 | ✓ | 2.9 | 1980-01 | 2011-08 | 31.7 (9.1) |
| Cimetta * | 1670 | 46.20042 | 8.79164 | 1995-12 | 2010-12 | 15.1 (8.4) | - | - | - |
| Davos | 1610 | 46.81297 | 9.84349 | 1999-01 | ✓ | 17.0 (13.4) | 1980-01 | 2005-11 | 25.9 (3.3) |
| Einsiedeln | 910 | 47.13304 | 8.75655 | 2012-03 | ✓ | 2.8 | 1980-01 | 2012-04 | 32.3 (9.7) |
| Elm | 958 | 46.92375 | 9.17534 | 2011-04 | ✓ | 3.7 | 1980-01 | ✓ | 35.0 (12.4) |
| Engelberg | 1035 | 46.82189 | 8.41044 | 2012-08 | ✓ | 2.4 | - | - | - |
| Fahy | 596 | 47.42382 | 6.94110 | 2009-11 | ✓ | 5.2 | - | - | - |
| Genève-Cointrin | 412 | 46.24751 | 6.12774 | 2012-05 | ✓ | 2.6 | 1980-01 | ✓ | 35.0 (12.4) |
| Glarus | 516 | 47.03458 | 9.06690 | 2013-08 | ✓ | 1.4 | - | - | - |
| Grächen | 1605 | 46.19531 | 7.83682 | 2013-06 | ✓ | 1.6 | 1980-01 | ✓ | 35.0 (12.4) |
| Grimsel Hospiz | 1980 | 46.57169 | 8.33325 | 2012-09 | ✓ | 2.3 | - | - | - |
| Gütsch ob And. | 2283 | 46.65244 | 8.61505 | 2005-09 | ✓ | 9.3 | - | - | - |
| Jungfraujoch | 3580 | 46.54745 | 7.98533 | 1999-01 | ✓ | 17.0 (13.4) | 1980-01 | ✓ | 35.0 (12.4) |
| Koppigen | 484 | 47.11884 | 7.60549 | 2012-01 | ✓ | 3.0 | 1980-01 | ✓ | 35.0 (12.4) |
| La Dôle | 1669 | 46.42470 | 6.09948 | 2009-10 | ✓ | 5.2 | - | - | - |
| Locarno * | 370 | 46.17223 | 8.78750 | 1995-12 | 2010-12 | 15.1 (8.4) | 1980-01 | ✓ | 35.0 (12.4) |
| Lugano | 273 | 46.00423 | 8.96031 | 2012-12 | ✓ | 2.1 | 1980-01 | ✓ | 35.0 (12.4) |
| Luzern | 454 | 47.03643 | 8.30096 | 2013-05 | ✓ | 1.7 | - | - | - |
| Magadino | 203 | 46.16003 | 8.93366 | 2006-02 | ✓ | 8.9 | 1980-01 | 2010-11 | 30.8 (12.4) |
| Napf | 1403 | 47.00466 | 7.94004 | 2007-07 | ✓ | 7.5 | - | - | - |
| Neuchâtel | 485 | 47.00006 | 6.95329 | 2010-10 | ✓ | 4.2 | - | - | - |
| Nyon | 455 | 46.40105 | 6.22775 | 2005-10 | 2009-07 | 3.7 | - | - | - |
| Payerne * | 490 | 46.81158 | 6.94242 | 1995-01 | 2010-12 | 16.0 (8.4) | 1980-01 | ✓ | 35.0 (12.4) |
| Plaffeien | 1042 | 46.74766 | 7.26600 | 2005-08 | 2009-06 | 3.9 | - | - | - |
| Poschiavo | 1078 | 46.34664 | 10.06113 | 2008-01 | ✓ | 6.9 | 1980-01 | ✓ | 35.0 (12.4) |
| Ruenenberg | 611 | 47.43456 | 7.87932 | 2013-12 | ✓ | 1.1 | - | - | - |
| Samedan | 1708 | 46.52640 | 9.87894 | 2012-12 | ✓ | 2.1 | 1980-01 | ✓ | 35.0 (12.4) |
| Schaffausen | 438 | 47.68977 | 8.62006 | 2008-08 | ✓ | 6.4 | 2004-02 | 2013-05 | 9.2 |
| Segl-Maria | 1804 | 46.43233 | 9.76230 | 2014-03 | ✓ | 0.8 | 1980-01 | 2014-06 | 34.5 (11.9) |
| Stabio | 353 | 45.84339 | 8.93238 | 2009-10 | ✓ | 5.2 | - | - | - |
| Ulrichen | 1345 | 46.50482 | 8.30814 | 2008-06 | ✓ | 6.6 | 1999-09 | ✓ | 15.3 (12.4) |
| Weissfluhjoch * | 2690 | 46.83334 | 9.80638 | 1994-09 | 2010-12 | 16.3 (8.4) | 1980-01 | 2008-06 | 28.5 (5.9) |
| Zürich Fluntern | 555 | 47.37792 | 8.56572 | 2012-10 | ✓ | 2.2 | 1980-01 | ✓ | 35.0 (12.4) |
| Zürich Kloten | 426 | 47.47961 | 8.53595 | 2010-03 | ✓ | 4.8 | 1980-01 | ✓ | 35.0 (12.4) |

*: Radiation data from the ASRB network. If not specified, radiation data are from MeteoSwiss. All SYNOP observations are from MeteoSwiss.

✓: End date of the ground-based record is after the end of the satellite records (2014-12-31).



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
