# Peer review of "Correction of CCI cloud data over the Swiss Alps using ground-based radiation measurements"

_Atmospheric Measurement Techniques, 2018_

## Referee Comment (RC1) · Anonymous Referee #1 · 20 Mar 2018

General comments:

The article is a study looking at the issue of false cloud detection over high altitude regions likely due to the presence of snow for the MODIS- and AVHRR-derived CLOUD_cci products. A cloud mask is generated using surface longwave and shortwave observations and compared against SYNOP and satellite masks. This data is used to create a decision-tree model with the intent of identifying false cloud detections in the satellite products. This is an important issue and the basic approach seems sound. The manuscript is understandable but there are several places where the phrasing is awkward or grammatical errors are present.
Directly comparing satellite observations with surface observations is difficult to do, and can result in large uncertainties. I think some more detail about how the comparison is being done and what the expected uncertainties are warranted. For example, are the authors using only near-nadir satellite measurements or are some of these comparisons occurring at the edge-of-scan? Is a satellite cloud shadow mask being used? Are there measurement times available from the SYNOP stations, or are they just assumed to be at 12 UTC?

The authors assess cloudiness as a function of altitude and season, but state that these relationships are likely caused by the well-known issue of detecting clouds over cold, bright surfaces (i.e. snow). I'm wondering why these comparisons don't include the use of a snow mask? Are there other factors being proposed as reasons for the false detections other than snow? Including a snow product might also give some insight as to how much of the issue is due to the satellite over-detection despite the known presence of snow versus the inadequacy of a coarse snow mask over the relatively fine spatial structure of mountainous regions.

I had difficulty assessing the skill of the model given the results shown. One issue was how the percentages were presented. For example in Page 20, Line 1: "In the Swiss Alps area, the use of the decision tree model as a quality filter permitted the rejection of 62 % of the false cloud detections in the satellite cloud property dataset, with the limitation of causing the removal of 7 % of real clouds in the process." – I believe in this sentence the 62% refers to a percentage of a percentage, while the 7% is an absolute cloud amount. This is a somewhat confusing way to present the results. This result refers to the last line of page 30: "Globally, $62\pm13$ % of the cloud mask overestimations are detected, reducing the systematic false positive error from $14.4\pm15.5$ to $4.3\pm2.8$ % but increasing the missed clouds from $8.7\pm3.5$ to $15.6\pm2.1$ %." – So the model reduces false positives by $\sim$10% and increases missed cloud by $\sim$7%, correct? If so I think this could be presented more clearly.

Specific comments:

The last line of the abstract says the systematic error is reduced to 4.3% with an increase of 7% missed clouds. Does this mean a 4.3% high bias after offsets from missed cloud and false detections?

Page 3, Line 11: NOAA-16 only used 1.6 during the day for a short part of its lifetime

P2, L5: the Heidinger reference is for PATMOS-x, which is not a Cloud_cci product

Figure 10. This is a nice graphic but it doesn't add much to the understanding of the model. It could probably be left out or changed to give more information.

Figure 11 shows much larger adjustments to pre-2002 AVHRR than MODIS. Drift seems to be a factor but the differences seem large even in the early part of the satellite record. Have you looked at the satellite-viewing angle over the Alps over the course of the record?

Table 2 seems to show that many of the false positives detections occur for optically thick, low clouds (e.g. Groups G and I), but these groups have some of the lowest False positives identified. I think the author's allude to this in the discussion on page 15, but it could be made clearer.

Technical correction

Abstract: Line 2: 'among others' should refer to something. 1st line in Intro is awkward Page 2, Line 17: This sentence is a run-on Page 2, Line 25: Is "uses as labels: a typo? Not sure what it means.

---

## Referee Comment (RC2) · Anonymous Referee #2 · 16 Apr 2018

**Comments on: Correction of CCI cloud data over the Swiss Alps using ground-based radiation measurements**

In this manuscript the authors present a method to correct cloud detection results over high altitude regions which are particularly challenging due to snow cover.

Overall, I believe the manuscript is generally well written and I really only have some minor comments and corrections. I therefore, recommend that it be considered for publication after minor revisions.

- **Page 2, Line 8:** These papers are still under review so 2017 needs to be changed to 2018.

- **Page 2, Line 17:** The author mentions shortwave and longwave measurements before mentioning passive. Also, microwave should be grouped with shortwave and longwave measurements. Finally, the next sentence uses "latter ones" which refers to a group that contains active instruments and passive microwave when I believe the author intends for this second sentence to refer only to active instruments.

  For example, this would be clearer "...passive measurements of shortwave, longwave and microwave, as well as active instruments such as cloud radars and lidars. The latter ones ..."

- **Page 3, Line 7:** 'open-source' refers to source code. Another terminology should be used.

- **Page 3, Line 8:** 'the National' → 'the US National'

- **Page 3, Line 9:** 'from MODIS' → 'from the MODIS'

- **Page 3, Line 9:** 'the National' → 'the US National'

- **Page 3, Line 12:** '3.7 µm. For' → '3.7 µm-channel. For'

- **Page 3, Line 18:** 'degrees' → 'degree'

- **Page 3, Line 18:** A more complete list: "data such as atmospheric pressure, temperature and ozone, snow and ice cover, and land and sea surface temperature, all coming from ECMWF ERA Interim (...) along with surface reflectance from the MODIS MCD43C1 product [Schaaf et al., 2011]."

- **Page 3, Line 18:** Choosing the phase based on cost is not how it is done for Cloud_cci. For all versions up to and including the version discussed in Stengel et al. [2017] a method based on that of Pavolonis and Heidinger [2004] and Pavolonis et al. [2005] was used. Newer versions use a neural network approach, much like the cloud mask.

  This brings up a very good question as cloud phase determination suffers from very similar problems as cloud detection. Have the authors thought about this in the context of high altitude regions?

- **Page 6, Line 15:** 'satellites' → "satellite's"

- **Page 9, Line 3:** 'pyrgeometers, respectively pyranometers,' → 'pyrgeometers and pyranometers, respectively,'

- **Page 9, Line 6:** I don't think the cosine of the incidence angle weighting is required for the thermal instrument.

- **Page 9, Line 13:** 'modified' → 'a modified'

- **Page 9, Line 13:** 'which have an' → 'which has an'

- **Page 9, Line 17:** 'come from' → 'comes from'

- **Page 9, Line 17:** Subscript 'sky' should not be italicized. Only subscripts that are variables should be italicized.

- **Page 9, Line 28:** Need a comma after the equation. And, 'sky' should not be italicized.

- **Page 9, Line 29:** 'approximated to 1' → 'approximated to unity'

- **Page 12, Line 11:** Need a comma after the equation. And, subscripts 'e' and 'm' should not italicized.

- **Page 12, Line 12:** Subscripts 'e' and 'm' should not italicized.

- **Page 12, Line 16:** Need a comma after the equation.

- **Page 14, Line 1:** Is their a formal way to train this decision tree that the author can describe and/or give references for?

- **Figure 10 caption:** 'letters corresponds to' → 'letters correspond to'

- **Page 17, Line 5:** 'twice larger' → 'twice as large'

- **Table A1 caption:** 'temporally with MODIS' → 'temporally with the MODIS'

**References**

Michael J. Pavolonis and Andrew K. Heidinger. Daytime cloud overlap detection from AVHRR and VIIRS. *Journal of Applied Meteorology*, 43(5):762–778, May 2004. doi:10.1175/2099.1.

Michael J. Pavolonis, Andrew K. Heidinger, and Taneil Uttal. Daytime global cloud typing from AVHRR and VIIRS: Algorithm description, validation, and comparisons. *Journal of Applied Meteorology*, 44(6):804–826, June 2005. doi:10.1175/JAM2236.1.

Crystal Barker Schaaf, Jichung Liu, Feng Gao, and Alan H. Strahle. *Aqua and Terra MODIS Albedo and Reflectance Anisotropy Products, In Land Remote Sensing and Global Environmental Change: NASA's Earch Observing System and the Science of ASTER and MODIS, Remote Sensing and Digital Image Processing Series*, volume 11, chapter 24, pages 549–561. Springer New York, 2011. doi:10.1007/978-1-4419-6749-7_24.

Martin Stengel, Stefan Stapelberg, Oliver Sus, Cornelia Schlundt, Caroline Poulsen, Gareth Thomas, Matthew Christensen, Cintia Carbajal Henken, Rene Preusker, Jürgen Fischer, Abhay Devasthale, Ulrika Willén, Karl-Göran Karlsson, Gregory R. McGarragh, Simon Proud, Adam C. Povey, Don G. Grainger, Jan Fokke Meirink, Artem Feofilov, Ralf Bennartz, Jedrezj Bojanowski, and Rainer Hollmann. Cloud property datasets retrieved from AVHRR, MODIS, AATSR and MERIS in the framework of the Cloud_cci project. *Earth System Science Data*, 9:881–904, 2017.

---

## Author Comment (AC1) · 25 May 2018

**Review of "Correction of CCI cloud data over the Swiss Alps using ground-based radiation measurements"**

doi:10.5194/amt-2018-18

**Authors' response to anonymous referee 1**

Fanny Jeanneret and co-authors
May 25th, 2018

We thank the reviewer for the review and presenting his/her suggestions. We have considered each one carefully, and answered below using these abbreviations:

**RC(n):** referee comment number n

**AR:** author's response

**AC:** author's changes in the manuscript (if appropriate)

**1    General comments**

**RC(1)** The manuscript is understandable but there are several places where the phrasing is awkward or grammatical errors are present.

**AR** The phrasing and grammar of the paper have been proof-read again. Changes (including corrections made following both referees' suggestions) can be observed in the marked-up manuscript version.

**RC(2)** Directly comparing satellite observations with surface observations is difficult to do, and can result in large uncertainties. I think some more detail about how the comparison is being done and what the expected uncertainties are warranted. For example, are the authors using only near-nadir satellite measurements or are some of these comparisons occurring at the edge-of-scan? Is a satellite cloud shadow mask being used?

**AR** There is no data quality flag available in the datasets to assess the reliability of the satellite cloud masks. Since no uncertainties are provided with SYNOP observations either, we could not quantify the comparisons uncertainty a posteriori. Only the expected uncertainties of the ground-based instruments are known, and detailed in Section 3.1.

Yes, some of the observations will include data taken at large scan angles, at the edges of the satellite image swaths.

There is no satellite cloud mask shadow, because that would only matter for land observations, not cloud observations.

**RC(3)** Are there measurement times available from the SYNOP stations, or are they just assumed to be at 12 UTC?

**AR** The exact SYNOP observation time was not available, this information has been added under Section 2.2.

**RC(4)** The authors assess cloudiness as a function of altitude and season, but state that these relationships are likely caused by the well-known issue of detecting clouds over cold, bright surfaces (i.e. snow). I'm wondering why these comparisons don't include the use of a snow mask? Are there other factors being proposed as reasons for the false detections other than snow? Including a snow product might also give some insight as to how much of the issue is due to the satellite over-detection despite the known presence of snow versus the inadequacy of a coarse snow mask over the relatively fine spatial structure of mountainous regions.

**AR** The reviewer is right, using a satellite snow mask could help for a better discrimination between snow-induced false detections and other sources of false detections. NASA products based on MODIS with 36 channels could be considered. A snow mask from CALIPSO/CALIOP is available but cannot be used since CALIPSO/CALIOP is already used to calibrate the artificial neural network producing the cloud mask in the Cloud_cci datasets. In any case, due to the limited project duration and funding, these additional analysis became out of the scope of this study.

**RC(5)** I had difficulty assessing the skill of the model given the results shown. One issue was how the percentages were presented. For example in Page 20, Line 1: "In the Swiss Alps area, the use of the decision tree model as a quality filter permitted the rejection of 62 % of the false cloud detections in the satellite cloud property dataset, with the limitation of causing the removal of 7 % of real clouds in the process." – I believe in this sentence the 62% refers to a percentage of a percentage, while the 7% is an absolute cloud amount. This is a somewhat confusing way to present the results. This result refers to the last line of page 14: "Globally, $62\pm13$ % of the cloud mask overestimations are detected, reducing the systematic false positive error from $14.4\pm15.5$ to $4.3\pm2.8$ % but increasing the missed clouds from $8.7\pm3.5$ to $15.6\pm2.1$ %." – So the model reduces false positives by $\sim10$ % and increases missed cloud by $\sim7$ %, correct? If so I think this could be presented more clearly.

**AR** The values are presented this way because it seems to be the easiest way to understand the results. As the amount of clouds also depends on the location used, using relative values is the simplest.

So, 62 % is relative to the amount of false positives in the dataset (62 % of the false clouds in the datasets are removed by the algorithm developed); 7 % is relative to the amount of clouds in the dataset (7 % of the clouds, which were truly clouds, are removed as well). Removing 62 % of the errors means that the systematic error in the dataset went down, from $14.4\pm15.5$ to $4.3\pm2.8$ %. The reviewer is correct, the error is reduced by $\sim10$ %. But the uncertainty also drastically reduces ($\pm15.5$ to $\pm2.8$), which means that someone using this algorithm can expect a lower amount of false clouds as well as a more homogeneous distribution of false clouds in space (smaller variations between different locations). Removing 7 % of the clouds to this extent is worth it for some applications, for instance for someone computing monthly means of cloud properties, as the cloud properties would be more accurate since not biased by values corresponding to retrievals performed in the absence of clouds. The impact of removing the false clouds above elevated locations is described in details in Section 4.3 and in Figure 11 (it is more obvious when comparing this figure to the time series in Stengel et al. [2017], Figure 9).

In conclusion, the model allows not only to reduce by 10 % the false positives, but also to decrease their uncertainty. This is not about simply shifting the problem, since the 7 % increase of missing clouds is global whereas the decrease is specific to areas with high amounts of false positives.

**AC** Section 4.2 and the abstract have been modified to better reflect this.

**2 Specific comments**

**RC(6)** The last line of the abstract says the systematic error is reduced to 4.3% with an increase of 7% missed clouds. Does this mean a 4.3% high bias after offsets from missed cloud and false detections?

**AR** We are not sure to understand what the reviewer meant in this comment, but think that our answer to the previous question may have clarified this subject.

**RC(7)** Page 3, Line 11: NOAA-16 only used 1.6 during the day for a short part of its lifetime

**AR** This has been clarified as follows:

**AC** "AVHRR on NOAA-16 had a different setup: a 1.6 $\mu$m channel was used in daytime instead of the 3.7 $\mu$m, until this was changed to the same setup as the others in May 2003."

**RC(8)** P2, L5: the Heidinger reference is for PATMOS-x, which is not a Cloud_cci product

**AR** Indeed, this was confusing and has been removed.

**RC(9)** Figure 10. This is a nice graphic but it doesn't add much to the understanding of the model. It could probably be left out or changed to give more information.

**AR** We thought about how to add more information (such as for instance, the percentage of well-classified points in each leaf of the tree) but then the figure gets huge, full of unnecessary numbers and difficult to understand. As it is now, the figure helps clarifying what the groups defined in Table 2 correspond to (as well as most of the text on page 15). Therefore, we would prefer to keep it this way.

**RC(10)** Figure 11 shows much larger adjustments to pre-2002 AVHRR than MODIS. Drift seems to be a factor but the differences seem large even in the early part of the satellite record. Have you looked at the satellite-viewing angle over the Alps over the course of the record?

**AR** As explained in Section 4.3, the model was trained only on MODIS data, which could count for some of the difference seen when applying the model on the AVHRR record. This is difficult to quantify. However, we mostly think that the time drifting is responsible as the pattern of differences pre-2002 (mostly on CFC and CPH in Figure 11 of the manuscript) largely reminds us of the time drifting pattern seen in Stengel et al. [2017], Figure 1 (added right below for simplicity). As can be observed, the time drifting is consequent, especially pre-2002, which seems to match particularly well with the much larger adjustments seen on AVHRR pre-2002, when compared to MODIS.

[Figure]

**Figure 1.** Overview of all sensors processed in Cloud_cci and their duration as a function of the daytime equator crossing time (AM: ante meridiem, before noon; PM: post meridiem, after noon). Sensors belonging to the same dataset are shown in the same color.

**RC(11)** Table 2 seems to show that many of the false positives detections occur for optically thick, low clouds (e.g. Groups G and I), but these groups have some of the lowest False positives identified. I think the author's allude to this in the discussion on page 15, but it could be made clearer.

**AR** Groups G and I indeed contain a lot of false clouds (17.8 and 19.4 %) when compared to the other groups. In these groups, the model has difficulties classifying these false clouds as such (only 12 and 10 % of them are identified). However, based on the height criteria we cannot classify these clouds as "low" clouds strictly speaking: these groups correspond to "clouds" (false or real ones) located above areas elevated of less than 3000m, with large COT values, and with cloud tops lower than 627 hPa ($\sim$ 4000 m) for group G and between 627 and 215 hPa ($\sim$ 4000 and 11'000 m) for group I. It is difficult to say anything more than the values presented in Table 2, or to draw some clear conclusions from these values. For this reason, our analysis stresses out that our model reaches its limits on such groups of points, and leaves to the reader drawing further conclusions (which are very difficult to prove).

**3   Technical corrections**

**RC(12)** Abstract: Line 2: 'among others' should refer to something.

**AR** Corrected

**RC(13)** 1st line in Intro is awkward

**AR** It has been shortened

**RC(14)** Page 2, Line 17: This sentence is a run-on

**AR** This has been changed following the suggestions of the other reviewer.

**RC(15)** Page 2, Line 25: Is "uses as labels: a typo? Not sure what it means.

**AR** Changed

**AC** "Once the ground-based cloud mask is computed [...], it is used as reference to train an automated algorithm [...]"

**References**

Martin Stengel, Stefan Stapelberg, Oliver Sus, Cornelia Schlundt, Caroline Poulsen, Gareth Thomas, Matthew Christensen, Cintia Carbajal Henken, Rene Preusker, Jürgen Fischer, Abhay Devasthale, Ulrika Willén, Karl-Göran Karlsson, Gregory R. McGarragh, Simon Proud, Adam C. Povey, Don G. Grainger, Jan Fokke Meirink, Artem Feofilov, Ralf Bennartz, Jedrzej Bojanowski, and Rainer Hollmann. 
[revised manuscript text omitted]

---

## Author Comment (AC2) · 25 May 2018

**Review of "Correction of CCI cloud data over the Swiss Alps using ground-based radiation measurements"**

doi:10.5194/amt-2018-18

**Authors' response to anonymous referee 2**

Fanny Jeanneret and co-authors
May 25th, 2018

We thank the reviewer for the review and presenting his/her suggestions. We have considered each one carefully, and answered below using these abbreviations:

**RC:** referee comment

**AR:** author's response

**AC:** author's changes in the manuscript (if appropriate)

**RC** Page 2, Line 8: These papers are still under review so 2017 needs to be changed to 2018.

**AR** Both are still under review and the DOIs correspond to the discussion papers (hence, 2017). The references will indeed be updated if the papers get approved before this one.

**RC** Page 2, Line 17: The author mentions shortwave and longwave measurements before mentioning passive. Also, microwave should be grouped with shortwave and longwave measurements. Finally, the next sentence uses 'latter ones' which refers to a group that contains active instruments and passive microwave when I believe the author intends for this second sentence to refer only to active instruments. For example, this would be clearer '. . . passive measurements of shortwave, longwave and microwave, as well as active instruments such as cloud radars and lidars. The latter ones ...'

**AR** Corrected, it is indeed simpler to understand now.

**RC** Page 3, Line 7: 'open-source' refers to source code. Another terminology should be used.

**AR** 'open-source' was replaced with 'open', here and in the introduction as well

**RC** Page 3, Line 8: 'the National' → 'the US National'

**AR** Corrected

**RC** Page 3, Line 9: 'from MODIS' → 'from the MODIS'

**AR** Corrected

**RC** Page 3, Line 9: 'the National' → 'the US National'

**AR** Corrected

**RC** Page 3, Line 12: '3.7 $\mu$m. For' → '3.7 $\mu$m-channel. For'

**AR** Corrected

**RC** Page 3, Line 26: A more complete list: "data such as atmospheric pressure, temperature and ozone, snow and ice cover, and land and sea surface temperature, all coming from ECMWF ERA Interim (...) along with surface reflectance from the MODIS MCD43C1 product (Schaaf et al., 2010)."

**AR** Corrected (see author's correction in the next question).

**RC** Page 3, Line 26: Choosing the phase based on cost is not how it is done for Cloud cci. For all versions up to and including the version discussed in Stengel et al. [2017] a method based on that of Pavolonis and Heidinger [2004] and Pavolonis et al. [2005] was used. Newer versions use a neural network approach, much like the cloud mask. This brings up a very good question as cloud phase determination suffers from very similar problems as cloud detection. Have the authors thought about this in the context of high altitude regions?

**AR** The algorithm described initially in our paper is indeed not the one producing the data used. We apologize and corrected this (see below). Indeed, the cloud phase is affected, similarly as all the retrieved microphysical variables are affected by the false cloud detection problem. Figure 5 shows, for each variable as well as for the cloud phase, the impact of false cloud detection on the retrieved microphysics.

**AC** "Cloud properties are retrieved from the satellite-measured radiances using an optimal estimation approach, following the theoretical basis for inverse retrieval methods described in Rodgers [2004]. The algorithm, called Community Cloud retrieval for Climate (CC4CL), works in three steps: first, a neural network trained on co-located data from CALIPSO-CALIOP [Winker et al., 2009] is run on the measured radiances to determine if a cloud is present in the retrieval scheme or not. Then, the cloud phase is determined with a decision tree, as proposed by Pavolonis and Heidinger [2004] and Pavolonis et al. [2005]. Lastly, the retrieval is done using the measured radiances and some ancillary data such as atmospheric pressure, temperature and ozone, snow and ice cover, and land and sea surface temperature, all coming from ECMWF ERA Interim [Dee et al., 2011], along with surface reflectance from the MODIS MCD43C1 product [Schaaf et al., 2010]. The cloud top pressure, cloud optical thickness and cloud effective radius are returned directly by the optimal estimation, whereas the cloud top height, cloud top temperature, cloud albedo, liquid and ice water path are then inferred from them. For more information, the algorithm is described in detail in Sus et al. [2017] and McGarragh et al. [2017]."

**RC** Page 6, Line 15: 'satellites' → "satellite's"

**AR** Corrected

**RC** Page 9, Line 3: 'pyrgeometers, respectively pyranometers,' → 'pyrgeometers and pyranometers, respectively,'

**AR** Corrected

**RC** Page 9, Line 6: I don't think the cosine of the incidence angle weighting is required for the thermal instrument.

**AR** A pyrgeometer's response (longwave) is weighted by the cosine of the incidence angle, as specified in the manual of the CGR4 instruments
(`http://www.kippzonen.com/Download/35/Instruction-Sheet-Pyrgeometers-CGR4`).
It is correct that it is far from being as important as for the pyranometers (shortwave).

**RC** Page 9, Line 13: 'modified' → 'a modified'
Page 9, Line 13: 'which have an' → 'which has an'

**AR** The sentence has been corrected this way:

**AC** "Older measurements from the ASRB network have been taken by modified Eppley PIR pyrgeometers, which have an observed uncertainty of 3 Wm$^{-2}$."

**RC** Page 9, Line 17: 'come from' → 'comes from'

**AR** Corrected

**RC** Page 9, Line 25: Subscript 'sky' should not be italicized. Only subscripts that are variables should be italicized.

**AR** This is indeed correct, subscripts have been checked and corrected everywhere.

**RC** Page 9, Line 28: Need a comma after the equation. And, 'sky' should not be italicized.

**AR** Corrected

**RC** Page 9, Line 29: 'approximated to 1' → 'approximated to unity'

**AR** Corrected

**RC** Page 12, Line 11: Need a comma after the equation. And, subscripts 'e' and 'm' should not italicized.

**AR** Corrected

**RC** Page 12, Line 12: Subscripts 'e' and 'm' should not italicized.

**AR** Corrected

**RC** Page 12, Line 16: Need a comma after the equation.

**AR** Corrected

**RC** Page 14, Line 1: Is their a formal way to train this decision tree that the author can describe and/or give references for?

**AR** The decision tree is trained using recursive partitioning (Breiman, 1984). The information and reference have been added to the manuscript.

**AC** "The training is done by 10-fold cross-validation with random sampling, using recursive partitioning as presented in Breiman (1984)."

**RC** Figure 10 caption: 'letters corresponds to' → 'letters correspond to'

**AR** Corrected

**RC** Page 17, Line 5: 'twice larger' → 'twice as large'

**AR** Corrected

**RC** Table A1 caption: 'temporally with MODIS' → 'temporally with the MODIS'

**AR** Corrected

**References**

Leo Breiman. *Classification and regression trees.* Chapman & Hall/CRC, New York, N.Y., 1984. ISBN 978-1-351-46049-1. URL `http://lib.myilibrary.com?id=1043565`. OCLC: 1022760542.

D. P. Dee, S. M. Uppala, A. J. Simmons, P. Berrisford, P. Poli, S. Kobayashi, U. Andrae, M. A. Balmaseda, G. Balsamo, P. Bauer, P. Bechtold, A. C. M. Beljaars, L. van de Berg, J. Bidlot, N. Bormann, C. Delsol, R. Dragani, M. Fuentes, A. J. Geer, L. Haimberger, S. B. Healy, H. Hersbach, E. V. Hólm, L. Isaksen, P. Kållberg, M. Köhler, M. Matricardi, A. P. McNally, B. M. Monge-Sanz, J.-J. Morcrette, B.-K. Park, C. Peubey, P. de Rosnay, C. Tavolato, J.-N. Thépaut, and F. Vitart. The ERA-Interim reanalysis: configuration and performance of the data assimilation system. *Quarterly Journal of the Royal Meteorological Society*, 137(656):553–597, April 2011. ISSN 00359009. doi: 10.1002/qj.828.

Gregory R. McGarragh, Caroline A. Poulsen, Gareth E. Thomas, Adam C. Povey, Oliver Sus, Stefan Stapelberg, Cornelia Schlundt, Simon Proud, Matthew W. Christensen, Martin Stengel, Rainer Hollmann, and Roy G. Grainger. The Community Cloud retrieval for CLimate (CC4cl). Part II: The optimal estimation approach. *Atmospheric Measurement Techniques Discussions*, pages 1–55, October 2017. ISSN 1867-8610. doi: 10.5194/amt-2017-333.

Michael J. Pavolonis and Andrew K. Heidinger. Daytime Cloud Overlap Detection from AVHRR and VIIRS. *Journal of Applied Meteorology*, 43(5):762–778, May 2004. ISSN 0894-8763, 1520-0450. doi: 10.1175/2099.1. URL `http://journals.ametsoc.org/doi/abs/10.1175/2099.1`.

Michael J. Pavolonis, Andrew K. Heidinger, and Taneil Uttal. Daytime Global Cloud Typing from AVHRR and VIIRS: Algorithm Description, Validation, and Comparisons. *Journal of Applied Meteorology*, 44(6): 804–826, June 2005. ISSN 0894-8763, 1520-0450. doi: 10.1175/JAM2236.1. URL `http://journals.ametsoc.org/doi/abs/10.1175/JAM2236.1`.

Clive D. Rodgers. *Inverse methods for atmospheric sounding: theory and practice.* Number 2 in Series on atmospheric oceanic and planetary physics. World Scientific, Singapore, reprinted edition, 2004. ISBN 978-981-02-2740-1.

Crystal Barker Schaaf, Jichung Liu, Feng Gao, and Alan H. Strahler. Aqua and Terra MODIS Albedo and Reflectance Anisotropy Products. In Bhaskar Ramachandran, Christopher O. Justice, and Michael J. Abrams, editors, *Land Remote Sensing and Global Environmental Change*, volume 11, pages 549–561. Springer New York, New York, NY, 2010. ISBN 978-1-4419-6748-0 978-1-4419-6749-7. doi: 10.1007/978-1-4419-6749-7_24. URL http://link.springer.com/10.1007/978-1-4419-6749-7_24.

Oliver Sus, Martin Stengel, Stefan Stapelberg, Gregory McGarragh, Caroline Poulsen, Adam C. Povey, Cornelia Schlundt, Gareth Thomas, Matthew Christensen, Simon Proud, Matthias Jerg, Roy Grainger, and Rainer Hollmann. The Community Cloud retrieval for Climate (CC4cl). Part I: A framework applied to multiple satellite imaging sensors. *Atmospheric Measurement Techniques Discussions*, pages 1–42, October 2017. ISSN 1867-8610. doi: 10.5194/amt-2017-334.

David M. Winker, Mark A. Vaughan, Ali Omar, Yongxiang Hu, Kathleen A. Powell, Zhaoyan Liu, William H. Hunt, and Stuart A. Young. Overview of the CALIPSO Mission and CALIOP Data Processing Algorithms. *Journal of Atmospheric and Oceanic Technology*, 26(11):2310–2323, November 2009. ISSN 0739-0572, 1520-0426. doi: 10.1175/2009JTECHA1281.1. URL http://journals.ametsoc.org/doi/abs/10.1175/2009JTECHA1281.1.

---

## Referee Report (RR1)

General comments:

The article is a study looking at the issue of false cloud detection over high altitude regions likely due to the presence of snow for the MODIS- and AVHRR-derived CLOUD_cci products. A cloud mask is generated using surface longwave and shortwave observations and compared against SYNOP and satellite masks. This data is used to create a decision-tree model with the intent of identifying false cloud detections in the satellite products. This is an important issue and the basic approach is sound. The manuscript has benefited from additional proof reading.

I appreciate the changes the authors have made to address my specific comments. I was disappointed that no significant changes were made to address my two larger concerns: the lack of a snow mask in the analysis and a more thorough analysis of the uncertainties inherent in validating satellite-based observations with surface measurements. That said I understand time and resource constraints, and on page 19 the authors acknowledge the issue of satellite viewing geometry and suggest it as a topic for further study.

I believe that the article provides a significant contribution in its current form and is suitable for publication.